# Residual Mechanical Properties and Durability of High-Strength Concrete with Polypropylene Fibers in High Temperatures

**DOI:** 10.3390/ma15134711

**Published:** 2022-07-05

**Authors:** Heron Freitas Resende, Elvys Dias Reis, Felipe Nascimento Arroyo, Matheus Henrique Morato de Moraes, Herisson Ferreira dos Santos, Enio Gomes da Silva, Francisco Antonio Rocco Lahr, Eduardo Chahud, Tulio Hallak Panzera, André Luis Christoforo, Luiz Antônio Melgaço Nunes Branco

**Affiliations:** 1Engineering School, Federal University of Minas Gerais, Belo Horizonte 31270-901, Minas Gerais, Brazil; heronfr@hotmail.com (H.F.R.); echahud@gmail.com (E.C.); luizmelg@fumec.br (L.A.M.N.B.); 2Engineering School, Federal Center for Technological Education of Minas Gerais, Belo Horizonte 30421-169, Minas Gerais, Brazil; elvysreis@yahoo.com.br; 3Department of Mechanical Engineering, Federal University of Sao Carlos, Sao Carlos 13565-905, Sao Paulo, Brazil; lipe.arroyo@gmail.com (F.N.A.); alchristoforo@ufscar.br (A.L.C.); 4Department of Engineering, Federal Institute of Rondonia, Ariquemes 76870-000, Rondonia, Brazil; herisson.santos@ifro.edu.br (H.F.d.S.); enio.gomes@ifro.edu.br (E.G.d.S.); 5São Carlos Engineering School, University of Sao Paulo, Sao Carlos 13566-590, Sao Paulo, Brazil; frocco@sc.usp.br; 6Department of Mechanical Engineering, Federal University of São João del-Rei, São João del-Rei 35701-970, Minas Gerais, Brazil; panzera@ufsj.edu.br

**Keywords:** cementitious materials, synthetic fibers, high temperatures, residual properties, regression models

## Abstract

Investigations into the fire resistance of high-strength concrete (HSC) is extremely important to optimize structural design in construction engineering. This work describes the influence of polypropylene fibers on the mechanical properties and durability of HSC at high temperatures (25, 100, 200, 400, 600 and 800 °C). HSC specimens with 2 kg/m^3^ composed of polypropylene fibers are tested in a temperature range of 25 to 800 °C, followed by microstructural analysis. In addition, a statistical analysis is designed to identify the effect of factors, namely temperature and polypropylene fibers, and their interactions on mechanical properties and water absorption, electrical resistivity, mass loss and ultrasonic velocity. Most of the properties are improved by the incorporation of fibers, obtaining highly predictable regression models. However, the polypropylene fibers reduce compressive strength but improve the residual mechanical properties up to 400 °C.

## 1. Introduction

In recent decades, the rapid development and increase in high-strength concrete (HSC) applications in civil construction have increased the risks of its exposure to high temperatures. This situation, which modifies its mechanical properties, also occurs in normal strength concrete (NSC), which loses approximately 25% and 75% of its strength when heated at temperatures to 300 and 600 °C, respectively [1,2,3,4,5]. Compared to NSC, HSC has better mechanical strength and durability, fundamental characteristics building performance. However, it is more prone to explosive spalling in fire situations due to its low porosity [6,7].

Researchers have studied and developed modern construction materials exposed to extreme conditions, such as argon for non-metallic materials at high temperatures [8], use of shungisite microfillers with polycarboxylate additives in cement compositions [9], application of micro-steel and propylene fibers applied to ultra-high performance (UHP) concrete for reliability analysis of crack impact resistance [10], and fiber-reinforced polymers (FRP) for reinforcing reinforced concrete structures subjected to high temperatures [11].

In this sense, incorporating polypropylene fibers into HSC may be the solution to the problem [12]. Synthetic fibers significantly affect the hydraulic behavior of fire-exposed concrete, as they form a network of small voids and permeable channels where pressurized steam can pass. In this way, it prevents the internal pressure from overcoming the concrete, minimizes the possibilities of HSC explosive splintering and extends the firefighting time and users’ evacuation [13,14,15,16]. However, there is no consensus on the appropriate dosage, size and type of polypropylene fibers in concrete. Hiremath and Yaragal [17] studied the performance of reactive powder concrete adding polypropylene fibers at levels between 0.1% and 0.9% to the cement mass, while Bentz [18] used between 0.2% and 0.5% of fibers in high-performance concrete (HPC). In this research, it was possible to obtain a 35% increase in the compressive strength of the developed concrete, besides avoiding the occurrence of spalling. Heo et al. [19] used polypropylene fibers with three diameters (0.012, 0.02 and 0.04 mm), three lengths (9, 12 and 19 mm) and contents of 0.05% and 0.2% in conventional concrete. It was evident that the fiber diameter did not affect the occurrence of spalling, and the length and melting point of fibers are found to be important parameters. Bilodeal et al. [12], however, investigated lightweight concretes with contents of 1.5 and 2.5 kg/m^3^ of multidimensional fibers up to 20 mm in length and 1.5, 2.5 and 3.5 kg/m^3^ of multifilament fibers up to 12.5 mm in length. It was possible to conclude that values close to 3.5 kg of the 20 mm polypropylene fibers per cubic meter of concrete is necessary to prevent the spalling of a low W/C lightweight concrete. Kalifa et al. [20] added 19 mm long polypropylene fibers to the high-performance concrete in grades of up to 3 kg/m^3^, concluding that the fiber dosage of 2 kg/m^3^ with a fiber length in the order of 20 mm is an efficient solution for preventing spalling in HPC up to 100 MPa under ISO 834 conditions. As it turns out, although these studies have addressed the influence of these factors on different types of concrete, extensive experimental research has not yet been carried out using the Eurocode 2 [3] recommendations in HSC [3,12,18,19,20,21].

In general, polypropylene fibers melt at approximately 170 °C, while spalling can occur in the temperature range between 190 and 250 °C. However, it is not yet consolidated in the literature what happens with the fibers after fusion. Kalifa et al. [20] suggest that the melted fibers are absorbed by the cement matrix, i.e., form a network of small voids available for vapor pressure release. Khoury [22], however, considers that the diffusion of synthetic fibers is impossible to occur in the cement matrix due to the fused fiber high viscosity. Therefore, it is suggested that the contact between the fibers and the cement matrix becomes permeable to water vapor before melting.

The remarkable application of HSC in structures leads to a better understanding of its behavior. Even if HSC has greater durability and mechanical resistance, fundamental characteristics for building performance, these are not enough to guarantee the useful life if this material is subject to a fire situation and if some essential care is not taken [2,5,23].

With this perspective (i.e., contribution to the high-temperature behavior of HSC when polypropylene fibers are added), this work analyzes the influence of high temperatures on the residual mechanical properties and the durability of polypropylene fiber-reinforced HSC. In this paper, after choosing the concrete recipe and checking the Spalling, seven properties were evaluated (i.e., compressive strength, tensile strength, elastic modulus, water absorption, electrical resistivity, mass loss and ultrasonic velocity). The incorporation of polypropylene fibers followed the dosage recommended by the Eurocode 2 [3] standard and used by Caetano et al. [24], equal to 2 kg/m^3^ of fibers. In the end, quadratic regression models were used to determine the properties as a function of temperature, aiming at an easy material characterization without the need for laboratory testing. Furthermore, microstructure images were used to evaluate fiber quality and behavior at high temperatures.

The present work aims to evaluate the influence of the addition of polypropylene fibers in HSC exposed to high temperatures, applying the methodology proposed by RILEM TC 129-MTH [25,26,27], verifying its consistency and applicability, as well as evaluating the HSC with and without polypropylene fibers at high temperatures (100, 200, 400, 600 and 800 °C) and determining the influence of fibers and temperature on their physical and mechanical properties such as compressive strength, modulus of elasticity, tensile strength, ultrasonic speed, electrical resistivity, water absorption, and mass loss. Simple regression models are also proposed that allow one to estimate the physical and mechanical properties as a function of temperature and the amount of polypropylene fibers applied to the HSC. Finally, qualitatively analyzing the microstructure of the samples to verify the influence of fiber melting on the microstructure is also proposed.

## 2. Materials and Methods

### 2.1. Materials

Portland Cement CPV-ARI RS (ASTM Type III with 3.10 specific gravity, 462 kg/m^2^ fineness and 135 and 205 min of start and end of setting time, respectively) [28] achieves high strengths in the first days of application. The influence of the cement type on the concrete mechanical properties at elevated temperatures is small [29,30,31]. The concrete was characterized by mechanical (i.e., compressive strength, tensile strength, and modulus of elasticity) and physical (i.e., water absorption, electrical resistivity, weight loss and ultrasonic speed) tests. The fine aggregate used has a specific gravity of 2632 kg/dm^3^, with a fineness modulus of 2855. The coarse aggregate has a specific gravity of 2694 kg/dm^3^ and a fineness modulus of 7029, respectively.

HSC was composed of natural sand and stone gneiss 1 with particle size distributions according to NBR 7211 standard [32]; potable water; polyfunctional water-reducing plasticizer additive, MIRA SET 278 (a specific gravity of 4.56 kg/m^3^, with an addition of 0.80% in relation to the cement); polyfunctional plasticizer additive—medium-range, MIRA FLOW 929 (a specific gravity of 4.56 kg/m^3^, with an addition of 0.80% in relation to the cement); micro silica; and polypropylene monofilament fibers (350 MPa of tensile strength) 12 mm long and 25 µm in diameter (Figure 1).

### 2.2. Mix Compositions, Specimen Preparation and Test Procedures

The concrete was produced at the Laboratory of Experimental Analysis of Structures (LAEEs) at the Federal University of Minas Gerais (UFMG). The mixture was carried out in a 320 L concrete mixer. All the inputs were inserted and mixed for about 15 to 18 min, the time necessary for the concrete to present adequate workability given the low water–cement ratio (w/c) value [33]. In addition, the specimens were molded in 2 or 3 steps and compacted on a vibrating table, as recommended by the RILEM TC 200-HTC standard [34].

Two batches of cylindrical specimens measuring 10 × 20 cm [34] were molded, one with 162 reference samples (without fibers) and another containing 162 samples with 2 kg/m^3^ of polypropylene fibers (0.35 wt% of the cement mass, according to Bilodeal et al. [12]). For the compressive and tensile strength tests, 9 specimens (18 total) were molded per temperature range. Since the other tests are non-destructive, 9 specimens more were needed per temperature range. The mix used was 1:1.4:1.4:0.35 (cement:sand:gravel:water) in both, leading to a consumption of 569.40 kg/m^3^ of cement, 796.80 kg/m^3^ of sand, 796.80 kg/m^3^ of gravel, 187.9 kg/m^3^ of water, 56.94 kg/m^3^ of micro silica, 4.56 kg/m^3^ of MIRA FLOW 929 additive and 4.59 kg/m^3^ of MIRA SET 278 additive. In the second batch, the fibers were dispersed in the dry HSC mixture; later, the concrete mixer turning ensured their complete dispersion in the mix. Then, water and additives were added to produce the HSC with fibers.

After 24 h of molding and being duly identified, the specimens remained in their molds, without humidity variation for the next 6 days, being covered to avoid water loss, according to the guidance of the RILEM TC 129-MTH recommendation [25,26,27]. In addition, they were stored in the humidity condition “d” (drying concrete) until completing at least 2 months, keeping the samples exposed to air in a climatized chamber (Figure 2) at a temperature of 20 ± 2 °C, with a relative humidity of 50 ± 5%. Finally, their extremities were mechanically treated, aiming to be flat and orthogonal to their central axis, aged between 28 days and 2 months.

The parameters studied in this paper were the mechanical properties and the durability of HSC with polypropylene fibers at room temperature (25 °C) and the residual mechanical properties and durability after exposure to high temperatures. The mechanical properties, such as compressive strength, elastic modulus and tensile strength, were obtained. In addition, durability, ultrasound, electrical resistivity, water absorption, mass loss and analysis of the possible occurrence of spalling were considered.

All specimens were exposed to 100, 200, 400, 600 and 800 °C using constant heating and cooling rates equal to 1 °C/min [25,26]. It was kept for one hour at the pre-set temperature. As recommended by RILEM TC 129-MTH [25,26,27], thermocouples were used (Figure 3a) since the oven used allows for the heating of two specimens (Figure 3b).

### 2.3. Statistical Analysis

The combination of temperature factors (T) (i.e., 25, 100, 200, 400, 600 and 800 °C) and polypropylene fibers (PF) (i.e., with and without) resulted in 12 different experimental treatments (Tr), as shown in Table 1.

Tukey’s mean contrast test, at a 5% significance level, was used to assess the influence of the individual factors (i.e., T, PF). From Tukey’s test, Letter A denotes that the factor level is effectively associated with the highest mean value, B with the second highest mean value and so on. Equal letters imply factor levels with equivalent means.

Analysis of variance (ANOVA), also considered at a 5% significance level, was used to assess the effects of main factors and their interactions (Table 1). A *p* value lower than the significance level (0.05) indicates the presence of significant effects. A quadratic regression model (Equation (1)) was used to estimate such properties.
(1)$$Y=\beta_{0}+\beta_{1}\cdot T+\beta_{2}\cdot T^{2}+\varepsilon$$

From Equation (1), *Y* denotes the property to be estimated (dependent variable), *T* (temperature) is the independent variable, *β_i_* are the model parameters to be adjusted by the least-squares method, and *ε* is the random error. The model’s quality (accuracy) was evaluated through the coefficient of determination (R^2^—Equation (2), wherein for *Y* data, *i* is the sample value of a property determined experimentally, $\bar{Y}$ data, *i* is the average value obtained from the *n* sample values obtained experimentally and $Y_{predict_{i}}$ is the property value estimated by the regression model). The closer to 100%, the better the developed equation’s accuracy. Equation (1) estimates values of the physical and mechanical properties of manufactured concrete for temperatures not considered in the experimental design but in the range between 25 and 800 °C.
(2)$$R^{2}(\%)=100\cdot \left(1-\frac{\sum_{i=1}^{n}{\left(Y_{predict_{i}}-Y_{data_{i}}\right)}^{2}}{\sum_{i=1}^{n}{\left(Y_{data_{i}}-{\bar{Y}}_{data_{i}}\right)}^{2}}\right)$$

To consider the interaction effect of factors (i.e., possibility of improving the adjustments precision), the quadratic regression model of two free variables (Equation (3)) was used. Regarding polypropylene fibers (PF), in Equation (3), 0 denotes concrete without polypropylene fibers, and 1 implies considering the fibers.
(3)$$Y=\beta_{0}+\beta_{1}\cdot T+\beta_{2}\cdot PF+\beta_{3}\cdot T^{2}+\beta_{4}\cdot T\cdot PF+\varepsilon$$

Based on the expressions obtained, it is possible to estimate the properties for values of fiber content equal to 0% or 0.35% (1) and temperature between 25 and 800 °C. Although some authors have proposed [35,36,37] representative models for the dynamic behavior of concrete with polypropylene fibers, when subjected to compression, no research was found in the literature involving models to estimate the mechanical and durability properties of fiber-reinforced HSC, even at room temperature.

### 2.4. Test Scheme

The compressive strength tests (Figure 4a) of the samples were carried out according to NBR 5739 [38] recommendations. EMIC hydraulic press PC 2000 CS model with 2000 kN capacity was used, at a loading rate of 0.45 ± 0.15 MPa/s. The equipment was calibrated according to NBR ISO 7500-1 [39]. To determine the elastic modulus in compression (Figure 4b), experiments were carried out in accordance with the recommendations of NBR 8255-1 [40], using the hydraulic press previously mentioned, at a loading rate of 0.45 ± 0.15 MPa/s. The recommendations of NBR 7222 [41] were followed to determine the tensile strength (Figure 4c), using the previously mentioned hydraulic press with a loading rate of 0.05 ± 0.01 MPa/s.

The experimental setups for the tests for compressive strength, modulus of elasticity in compression, and tensile strength are presented in Figure 4 and followed the recommendations of NBR 5739 [38], NBR 8255-1 [40], and NBR 7222 [41].

## 3. Results

Through the material characterization, the compressive strengths obtained at 1, 3, 7 and 28 days were 19.6, 31.8, 41.3 and 52.4 MPa, meeting the Brazilian standard NBR 5739 [38] requirements. The specific area (Blaine) obtained was 462 m^2^/kg, following the NBR 16,372 standard [42], which recommends a value equal to or greater than 300 m^2^/kg. The start and end times recorded were 135 and 205 min, according to the NBR 16,607 standard [43]. Furthermore, the density of the samples obtained was 0.91 kg/m^3^.

### 3.1. HSC Performance at High Temperatures

For each temperature (i.e., 25, 100, 200, 400, 600 and 800 °C), a specimen of HSC-Ref (i.e., without fibers) and another of HSC-PF (i.e., with fibers) were tested after three months of curing in air. Spalling did not occur in any of the trials, which means that the methodology used is consistent. A similar method was reported by Amancio et al. [44], which did not register spalling up to 600 °C.

### 3.2. Sample Surface Physical Observation

Figure 5a,b shows the surface of HSC-Ref and HSC-PF specimens, respectively, exposed to pre-established temperatures. At 100 and 200 °C, there was no color change and no visible cracks on the HSC-Ref and HSC-PF sample surfaces. At 400 °C, microcracks were present, and the color changed to a darker gray on the HSC-Ref sample surface, while the HSC-PF sample showed no superficial cracks, but a slight color change to a darker gray. At 600 °C, both samples showed visible cracks, but the HSC-Ref was more evident, with small pores and color change to a lighter gray. At 800 °C, the development of cracks and voids on the surface was more pronounced, especially for the HSC-Ref sample, and there was no color change in any specimen. Furthermore, there was no fragmentation of the HSC-PF, indicating that the use of polypropylene fibers is sufficient to prevent explosive fragmentation up to 800 °C, validating the methodology adopted in the heating tests.

These findings indicate that using polypropylene fibers in HSC reduces surface pores and cracks. This is attributed to the fiber melting and vaporizing, considering their lower melting point, creating microchannels in the concrete. Thus, a higher vapor tension in the capillaries can be relieved and released, explaining the non-occurrence of spalling [45].

### 3.3. Physical and Mechanical Properties Result

In the following sections, all figures “a” and “b” present the results of Tukey’s test for the samples without and with polypropylene fibers, respectively. Thus, it will be possible to judge if the temperature increase affected the results. To evaluate the addition of polypropylene fibers, figure “c” was prepared. Through this, it will be possible to analyze if, at each temperature level, which sample obtained the best results or if they are equal, i.e., if the addition of fibers affected the results at these levels. Finally, figure “d” shows the quadratic regression model fit to the experimental data set considering only the effect of temperature as an independent factor. Furthermore, the error bars (standard deviation) and the minimum and maximum values of the coefficient of variation (CV) are presented.

#### 3.3.1. Compressive Strength (CS)

Figure 6 below shows the results for the compressive strength test. All statistical analyses were validated by ANOVA.

The HSC-Ref reduces strength by up to 62.66% by increasing the temperature level. The HSC-PF provides a slight reduction of 64.12%. However, at 100 °C, an increase in strength can be noted when the fibers are incorporated, while HSC-Ref maintains the same strength. This occurs because the polypropylene fibers melt at a temperature below 300 °C, increasing the HSC porosity, including more escape routes, with a consequent reduction in the water vapor pressure. However, the fiber melting causes thermal incompatibility between the aggregate and the cement paste, increasing the free space and creating a thermal buffer. Thus, it can be said that the fusion of synthetic fibers favors water evaporation and improves the CS up to a temperature of 200 °C.

As indicated in the literature, synthetic fibers do not significantly impact the compressive strength of unheated HSC [46]. The same can be observed in this study, in which, for a temperature of 25 °C, the resistances are equivalent. When analyzing the other temperature ranges, up to 600 °C, the HSC-PF has superior resistance to the HSC-Ref. However, at 800 °C, the HSC-Ref has a superior resistance.

According to the literature, the results of residual compressive strength of fiber-reinforced HSCs are contradictory, indicating reductions [47,48] and other increases [20,48]. This can be attributed to the experimental conditions such as curing, sample conditions and heating rate. Amancio et al. [44] indicated an interaction between fiber and temperature only for compressive strengths of HSC from 600 °C.

Finally, analyzing the regression model, one can notice an excellent precision in the proposed model (R^2^ = 98.02%) [49]. Therefore, it is possible to estimate the mass loss for temperatures between 25 and 800 °C, in addition to discarding the need for other experimental tests and generating higher costs.

The compressive strength values estimation based on temperatures and fiber contents is presented below (Equation (4)). The adjustment is illustrated in Figure 7.
(4)$$CS=57.92-0.02456\cdot T+7.52\cdot PF-0.000027\cdot T^{2}-0.01129\cdot T\cdot PF\;R^{2}=93.17\%$$

It is noted that the combination of two factors ended up reducing the coefficient of determination value when compared with the quadratic model, dependent only on temperature (i.e., variability of polypropylene fibers not explained by the equation). However, this model also has high predictability (R^2^ = 93.17%) [49]. Through this model, it is possible to analyze how other percentages of polypropylene fibers would affect the results obtained. From this graph, 1 represents the incorporation of 2 kg/m^3^, while 0.5 represents the inclusion of 1 kg/m^3^.

#### 3.3.2. Elastic Modulus in Compression (E)

Figure 8 below shows the results for the elastic modulus in the compression test. All statistical analyses were validated by ANOVA.

The modulus of elasticity can reduce up to 86% by increasing temperature, with or without fibers. However, for HSC-Ref, the reduction started from 400 °C, while fiber-reinforced HSC reduced from 200 °C. This can be attributed to a decrease in internal vapor pressure through the channel connections generated by the melted polypropylene fibers, such that this pressure can be evacuated more efficiently [50]. Unlike for compressive strength, the modulus of elasticity can be considered similar in all temperature levels, except at 200 °C, in which the HSC-Ref has a higher modulus.

This result agrees with some studies in the literature, which indicate that the modulus of elasticity is greater at 25 °C [51,52]. Therefore, it is inferred that stiffness is decreased by increasing temperature due to the microcracks in the cement paste caused by the thermal cycle in the heating step and the additional effect on the cooling process [50]. This property can be affected by the same factors that influence its compressive strength [53].

Finally, analyzing the regression model, one can notice an excellent precision in the proposed model (R^2^ = 91.41%) [49]. Therefore, it is possible to estimate the mass loss for temperatures between 25 and 800 °C, in addition to discarding the need for other experimental tests and generating higher costs.

The elastic modulus values estimation based on temperatures and fiber contents is presented below (Equation (5)). The adjustment is illustrated in Figure 9.
(5)$$E=33.805-0.05310\cdot T-0.746\cdot PF+0.000019\cdot T^{2}+0.00116\cdot T\cdot PF\;R^{2}=96.15\%$$

Based on the results of Equation (5), it can be seen that the combination of two factors ended up increasing the coefficient of determination value when compared to the quadratic model depending only on temperature (Figure 8d). Thus, the regression model of Equation (5) is considered the most suitable for estimating the modulus of elasticity of concrete with and without fibers. Furthermore, as explained in Figure 7, one can analyze how other percentages of polypropylene fibers would affect the results obtained.

#### 3.3.3. Tensile Strength (TS)

Figure 10 below shows the results for the tensile strength test. All statistical analyses were validated by ANOVA.

As can be seen, both HSC-Ref and HSC-PF do not reduce the tensile strength up to 100 °C. However, at 200 °C, while HSC-Ref reduces strength, HSC-PF increases it. The same occurred in the compressive strength (the increase started at 100 and 200 °C), which can be explained mainly by the development of secondary hydrates and the conversion of the remaining non-hydrated cement grains. These grains hydrate rapidly and produce a secondary hydrated gel with the participation of micro silica at this temperature. This can also occur due to adequate bonding between the fiber and the matrix, which indicates a positive influence of the polypropylene fibers on tensile strength and HSC enhancement up to 200 °C. Another reason for the increase is the autoclave effect and the creation of stronger siloxane elements, which cause an increase in strength at temperature levels between 250 and 350 °C [54]. After 200 °C, both HSCs lost strength.

When analyzed by temperature range, it can be noted that the HSC-PF has greater resistance when compared to the HSC-Ref, except in the range between 400 and 600 °C. The same can be confirmed in the literature [55]. This substantial reduction in HSC-PF samples is mainly attributed to the dense network of melted channels created by the evaporation of polypropylene fibers under high temperatures. These melted channels accumulate in a single location, initiating internal cracks that result in a sudden sample failure with fibers under tensile load [45]. Another possible cause is that as the temperature increases, the tetrahedral chains of quartz molecules elongate and reorient themselves. This leads to a significant increase in volume, which causes radial cracks around the heated sample particle perimeter [56].

Finally, analyzing the regression model, one can notice an excellent precision in the proposed model (R^2^ = 99.27%) [49]. Therefore, it is possible to estimate the mass loss for temperatures between 25 and 800 °C, in addition to discarding the need for other experimental tests and generating higher costs.

The estimation of tensile strength values based on temperature and fiber content levels is presented below (Equation (6)). The adjustment is illustrated in Figure 11.
(6)$$TS=4.934-0.003557\cdot T+0.564\cdot PF-0.000002\cdot T^{2}-0.00076\cdot T\cdot PF\;R^{2}=94.41\%$$

Based on the results of Equation (6), it can be seen that the combination of two factors ended up reducing the coefficient of determination value when compared to the quadratic model depending only on temperature (Figure 10d). Thus, the regression model in Figure 9d is considered the most suitable for estimating the modulus of elasticity of concrete with and without fibers. However, this model also has high quality (R^2^ = 94.41%) [49] and can be used to analyze how other percentages of polypropylene fibers would affect the results obtained.

#### 3.3.4. Absorption (Abs)

Figure 12 below shows the results for the absorption test. All statistical analyses were validated by ANOVA.

The temperature progressively increased the water absorption of both HSC, which increased by 3440% and 2243% for the HSC-Ref and HSC-PF, respectively. The most prominent point is the temperature transition from 100 to 200 °C, in which there was substantial growth in all samples. The same trend was observed at 400 °C. This increase can be attributed to the greater penetration of water through the voids and channels created by the melted polypropylene fibers [45]. At 600 °C, the increasing trend was repeated, revealing the influence of melted fibers and the presence of channels, as discussed in Section 3.1.

At all temperature levels, the unreinforced concrete had lower absorption when compared to the fiber-reinforced concrete results. In addition to the previous discussions, another reason for these results may be the effect of polypropylene fibers on workability, as they lead to decreased fluidity and additional pores in the concrete mix [57]. It is also noted that all fiber-reinforced concretes have higher mean values than the reference samples for the same temperature levels, except at 25 °C where the absorption is the same.

In this context, Kalifa et al. [20] attributed the increase in porosity and, consequently, the absorption rates by the increased temperatures, to the escape of water and the micro-cracks generated by expansions between the aggregate and the cement paste. However, Fares et al. [58] associated the evolution of these parameters to the output of water absorbed in the capillary pores and the release of water in the cement paste hydration products. Ye et al. [59] indicated that this increase is due to hydrated calcium silicate and calcium hydroxide decomposition.

Finally, analyzing the regression model, one can notice an excellent precision in the proposed model (R^2^ = 97.27%) [49]. Therefore, it is possible to estimate the mass loss for temperatures between 25 and 800 °C, in addition to discarding the need for other experimental tests and generating higher costs.

Below is the estimation of absorption values based on temperatures and fiber contents (Equation (6)). The adjustment is illustrated in Figure 13.
(7)$$Abs=-0.640+0.026011\cdot T+0.333\cdot PF-0.000017\cdot T^{2}+0.000618\cdot T\cdot PF\;R^{2}=97.69\%$$

Given the results of Equation (6), it is noted that the combination of two factors ended up increasing, even if only slightly, for the coefficient of determination value when compared with the quadratic model depending only on temperature (Figure 12d). Therefore, it is understood that both regression models are suitable for estimating the Abs of concrete with and without the presence of synthetic fibers.

#### 3.3.5. Electrical Resistivity (ER)

Figure 14 below shows the results for the electrical resistivity test. All statistical analyses were validated by ANOVA.

Electrical resistivity is a qualitative parameter of durability characterized by the movement of ions in the pore network and is directly linked to the moisture contained in the structure [60]. This parameter indicates the degree of difficulty in the passage of electric current through the material, representing the concrete’s ability to resist the conduction of ions in its structure [61]. Based on the RE average variation in the concretes with and without polypropylene fibers between temperatures of 25 and 800 °C, the risk of reinforcements corrosion can then be evaluated, as shown in Table 2.

The electrical resistivity affected HSC-Ref more than HSC-PF, since it presented higher values in all the analyzed temperature ranges. Up to 200 °C, HSC-PF maintained the same electrical resistivity, characterized by a negligible risk of reinforcement corrosion. For the HSC-Ref, this value was reduced, although it is still in the insignificant corrosion range. Both concretes, from 400 °C, have a low corrosion risk and a high and very high corrosion risk in the range of 600 and 800 °C, respectively. At 400 °C, there was a marked increase in RE loss for all samples at room temperature, 37.19% for HSC-Ref and 31.21% for HSC-PF. In this case, HSC-PF performed better by releasing the concrete core vapor pressure, providing good capacity to resist ion conduction and lower moisture loss, as had occurred at 200 °C.

It is understood that the HSC-PF sample absorbs smaller amounts of water in the specimens’ molding process since the polypropylene fibers occupy more of the void spaces in the pores. This explains the reduction (18.70%) in the concrete ER after the inclusion of fibers. The resistivity difference between the samples is due to the insufficiency of HSC-Ref to release the concrete internal vapor pressure [50]. This possibly reduces the ability to resist the conduction of ions in its structure and provides a greater loss of moisture and, consequently, electrical resistivity.

Finally, analyzing the regression model, one can notice an excellent precision in the proposed model (R^2^ = 98.60%) [49]. Therefore, it is possible to estimate the mass loss for temperatures between 25 and 800 °C, in addition to discarding the need for other experimental tests and generating higher costs.

The estimate of electrical resistivity values based on temperatures and fiber contents is presented below (Equation (8)). The adjustment is illustrated in Figure 15.
(8)$$ER=302.65-0.2541\cdot T-36.11\cdot PF-0.000106\cdot T^{2}+0.0313\cdot T\cdot PF\;R^{2}=97.08\%$$

Based on the results of Equation (8), it is noted that the combination of two factors ended up reducing, even if only slightly, the coefficient of determination value, when compared to the quadratic model, depending only on temperature (Figure 14d). Therefore, it is understood that both regression models are suitable for estimating the electrical resistivity of concrete with and without the presence of synthetic fibers.

#### 3.3.6. Weight Loss (WL)

Figure 16 below shows the results for the weight loss test. All statistical analyses were validated by ANOVA.

Before discussing the results, mass losses are more pronounced at temperatures above 100 °C. Below this value, there was no mass loss in any sample. There are many potential causes for such loss after concrete exposure to high temperatures. However, lump ejections or chipping of material surface layers tend to be the main reasons [63].

As can be seen, there was a low dispersion of results around the mean (highest CV equal to 1.34). Furthermore, the increase in temperature leads to greater mass loss in all samples, with the HSC-PF losses always being higher than those of HSC-Ref. At temperatures above 600 °C, the polypropylene fibers pass from the molten state and evaporate, increasing the HSC-PF sample weight loss compared to the HSC-Ref sample. This loss occurs mainly due to the water loss in three forms: free water, adsorbed water, and chemically combined water. Therefore, it is understood that the presence of fibers increases mass losses, as reported in the literature [64].

Finally, analyzing the regression model, one can notice an excellent precision in the proposed model (R^2^ = 96.04%) [49]. Therefore, it is possible to estimate the mass loss for temperatures between 25 and 800 °C, in addition to discarding the need for other experimental tests and generating higher costs. Below is the estimate of electrical resistivity values based on temperatures and fiber contents (Equation (9)). The adjustment is illustrated in Figure 17.
(9)$$WL=1.204-0.130\cdot T+0.01355\cdot PF-0.000007\cdot T^{2}+0.001885\cdot T\cdot PF\;R^{2}=97.55\%$$

It is noted that the combination of two factors ended up increasing, even if only slightly, the coefficient of determination value, when compared with the quadratic model, depending only on temperature (Figure 16d). Therefore, it is understood that both regression models are suitable for estimating the electrical resistivity of concrete with and without the presence of synthetic fibers.

#### 3.3.7. Ultrasonic Speed (US)

Figure 18 below shows the results for the ultrasonic speed test. All statistical analyses were validated by ANOVA.

The test used for US determination consists of a technique sensitive to degradation phenomena, including internal cracks and other alterations due to treatment. Thus, the results can be analyzed by correlating US and concrete quality (Table 3).

A low dispersion around the mean (higher CV equal to 3.65) was evidenced. The increase in temperature leads to reduced US values for all samples, regardless of the presence or not of fibers. Furthermore, the rise in temperature leads to lower ultrasonic velocity in all samples, which means a decrease in their quality. In all temperature ranges, except for 200 °C, HSC-Ref has better quality. For 800 °C, both have the same quality. From 100 °C, HSC-PF changes from excellent quality to very good quality, while HSC-Ref maintains excellent quality. Both have a regular quality in the range of 400 °C and bad from 600 °C.

The greater number of cracks delays the pulse velocity in the concrete [63], resulting in low US values. Thus, it is understood that the HSC-PF quality reduces due to evaporation of the polypropylene fibers, creating channels that increase the internal microcracks. As a result of this increase, the quality of the concrete decreases. From 600 °C, the samples of both concretes suffer thermal incompatibility between the cement paste and the aggregate, altering the concrete mechanical properties. Yang et al. [68] found results similar to those mentioned above in their experiments.

Finally, analyzing the regression model, one can notice an excellent precision in the proposed model (R^2^ = 98.03%) [49]. Thus, it is possible to estimate the ultrasonic velocity for temperatures between 25 and 800 °C, in addition to discarding the need for other experimental tests and generating higher costs.

Next, the ultrasonic velocity values estimated based on temperatures and fiber contents are presented in Equation (10). The adjustment is illustrated in Figure 19.
(10)$$US=5117.1-7.785\cdot T-73.40\cdot PF+0.003132\cdot T^{2}+0.067\cdot T\cdot PF\;R^{2}=97.72\%$$

The results from Equation (10) reveal that the combination of two factors ended up reducing, even if only slightly, the coefficient of determination value when compared with the quadratic model, which depends only on temperature (Figure 18d). Therefore, it is understood that both regression models are suitable for estimating the ultrasonic velocity of concrete with and without the presence of synthetic fibers.

### 3.4. Microstructure Analysis

The microstructure images were performed to make a qualitative durability analysis of HSC with and without polypropylene fibers. The microstructures of HSC at 25, 100 and 200 °C are shown in Figure 20. For this study, a microscope with a maximum magnification capability of 70 times was used.

Figure 20 shows a dense microstructure with closed arrays of hydrated compounds in both HSC-Ref and HSC-PF, with the polypropylene fibers closely bonded to the cement paste at 25, 100 and 200 °C. The increase in compressive strength concerning its value at room temperature was observed for the HSC-PF samples at 100 and 200 °C.

At 400 °C, the compressive strength of HSC-PF reduced more than HSC-Ref because the polypropylene fibers melted, affecting its microstructure and creating channels as indicated by Figure 21b. In addition, the concrete porosity significantly impacts the pore’s vapor pressure and the fusion of synthetic fibers, which benefits the water evaporation and improves the HSC-PF strength up to a temperature of 200 °C, as explained before.

At 600 °C, the HSC-Ref and HSC-PF microstructure reveals that the decomposition of calcium hydroxide and a considerable number of cracks occur due to the cement paste thermal expansion, which causes a local breakdown of the bond between cement and aggregate. Consequently, a high reduction in compressive strength was observed in the HSC-Ref and HSC-PF samples, with similar values for this temperature level. The channels generated by fiber melting and the cracks developed as a result of thermal expansion were observed in the images obtained using a magnifying glass at 600 °C, as shown in Figure 21d.

As the temperature increases, concretes with and without fibers deteriorate continuously. At 800 °C, there is an increase in porosity and a weak interface zone between the aggregate and cement paste, reducing the compressive strength of the HSC. The HSC-PF strength at 800 °C was slightly lower than HSC-Ref at this temperature threshold. This was due to the multiple channels of melted polypropylene fibers and cracks at the melt channel boundaries, as shown in Figure 21f. At this temperature, the HSC microstructure disintegrates with rough grains, resulting in very low residual compressive strength.

## 4. Conclusions

In this study, the influence of high temperatures on the residual mechanical properties and durability of high-strength concrete (HSC) with the addition of polypropylene fibers was investigated. Following, important conclusions are drawn from this study:It is understood that the methodology proposed by RILEM TC 129-MTH [25,26,27] and used in this paper is consistent, since spalling did not occur.The addition of polypropylene fibers to unheated HSC significantly reduces its compressive strength but improves its residual mechanical properties up to 400 °C.As expected, the presence of polypropylene fibers increased the compressive modulus (E) of HSC only at room temperature. However, the increase in temperature up to 800 °C did not change the reduction trend in the value of this property.The tensile strength (TS) of HSC increased when polypropylene fibers were added to the cement matrix at room temperature (i.e., at 100 and 200 °C), with the most significant gain being in the latter level (i.e., 200 °C). Above 400 °C, the use of fibers in the HSC reduces its tensile strength.The absorption (Abs) of HSC increased by increasing temperature regardless of the presence or not of polypropylene fibers. HSC-PF always had higher Abs than HSC-Ref, which is due to the greater penetration of water through the channels created by the fiber melting.HSC-Ref has higher electrical resistivity (ER) than HSC-PF at all tested temperature levels (25 to 800 °C). However, polypropylene fibers improve the HSC-PF performance up to 400 °C. Regardless of the use or not of fibers, RE reduces at high temperatures, indicating an increased risk of reinforcement corrosion.There was no mass loss (WL) at 100 °C, but between 200 and 800 °C, the use of polypropylene fibers in HSC worsened its performance, leading to increasing mass losses compared to unreinforced concrete.The ultrasonic velocity (US) values decreased as the temperature increased, regardless of whether polypropylene fibers were added to HSC. Even so, HSC-PF performed better than HSC-Ref up to 200 °C, demonstrating the positive effect of the fibers for this property. Above 400 °C, there was a high reduction in US and similar values for all samples, indicating poor quality of HSC.The qualitative microstructure analysis of the samples showed that in HSC-PF, the fusion of synthetic fibers benefits the evaporation of water and improves the compressive strength up to 200 °C. In contrast, at high temperatures, channels created by the fibers directly melting affect their microstructure, causing a substantial reduction of this property.The simple regression models showed excellent fits (R^2^ > 91.41%), i.e., it is possible to use temperature to estimate the investigated properties. Furthermore, the multiple regression models were also significant, i.e., it is possible to use PF in property estimation.

## Figures and Tables

**Figure 1 materials-15-04711-f001:**
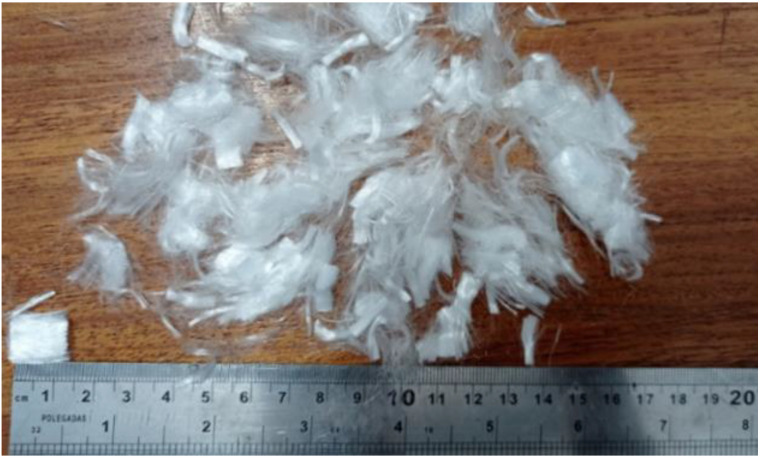
Polypropylene fibers supplied by Viapol.

**Figure 2 materials-15-04711-f002:**
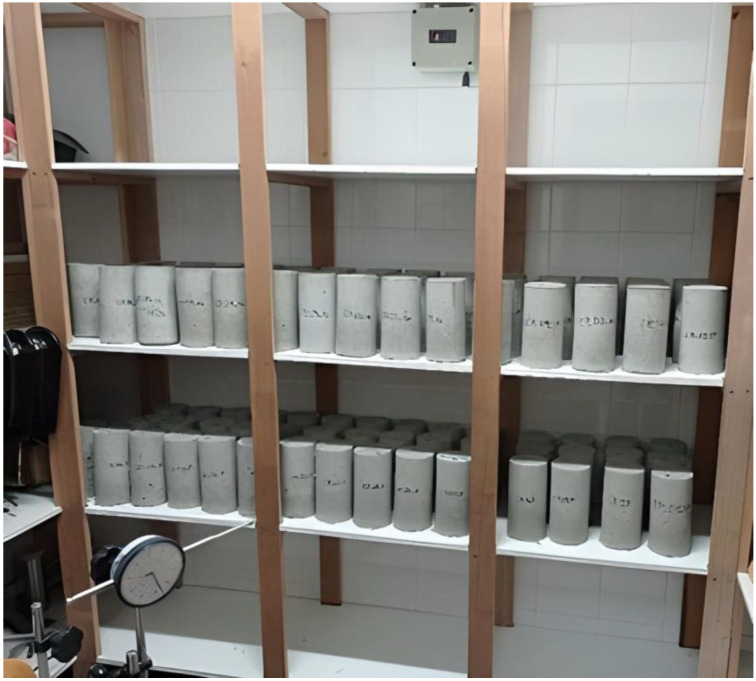
Mechanically treated ends’ specimens stored in the climatized chamber.

**Figure 3 materials-15-04711-f003:**
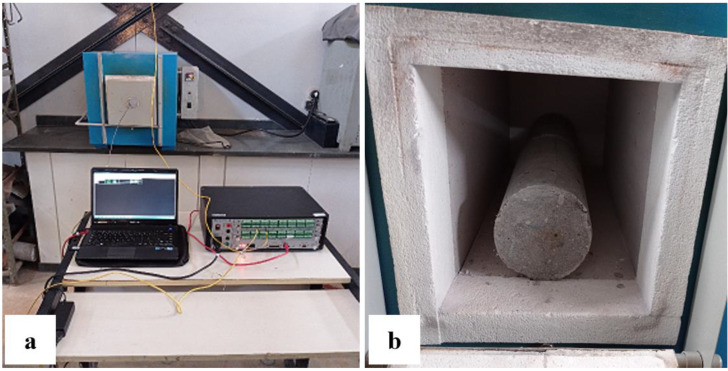
Heating test: (**a**) thermocouple and (**b**) specimens inside the furnace.

**Figure 4 materials-15-04711-f004:**
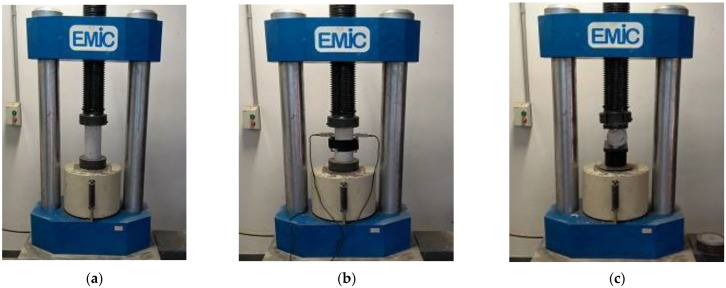
Test scheme: (**a**) compressive strength, (**b**) modulus of elasticity in compression, and (**c**) tensile strength.

**Figure 5 materials-15-04711-f005:**
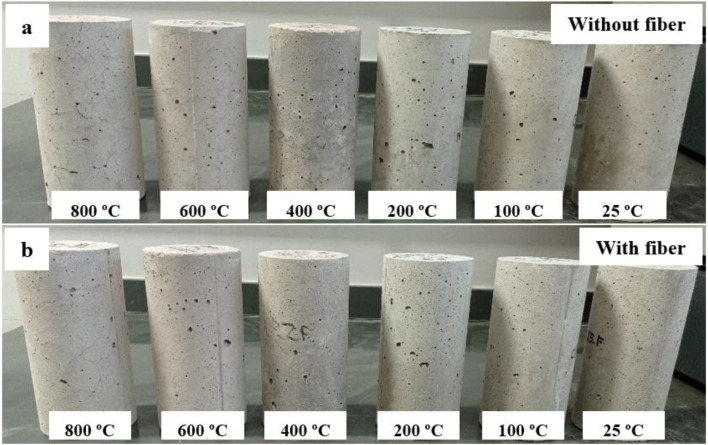
Specimen surfaces after heating test: (**a**) HSC-Ref and (**b**) HSC-PF.

**Figure 6 materials-15-04711-f006:**
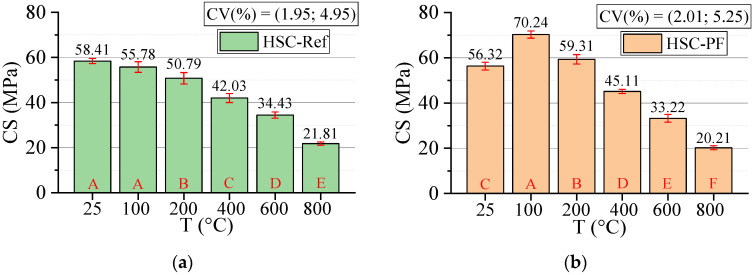
Compressive strength (CS) results summary: (**a**) Tukey’s test for the samples without polypropylene fibers, (**b**) Tukey’s test for the samples with polypropylene fibers, (**c**) Tukey’s test for the samples without and with polypropylene fibers and (**d**) Quadratic regression model fit to the experimental data set considering only the effect of temperature as an independent factor. The uppercase letters A–F in figure mean equivalence between the sample results.

**Figure 7 materials-15-04711-f007:**
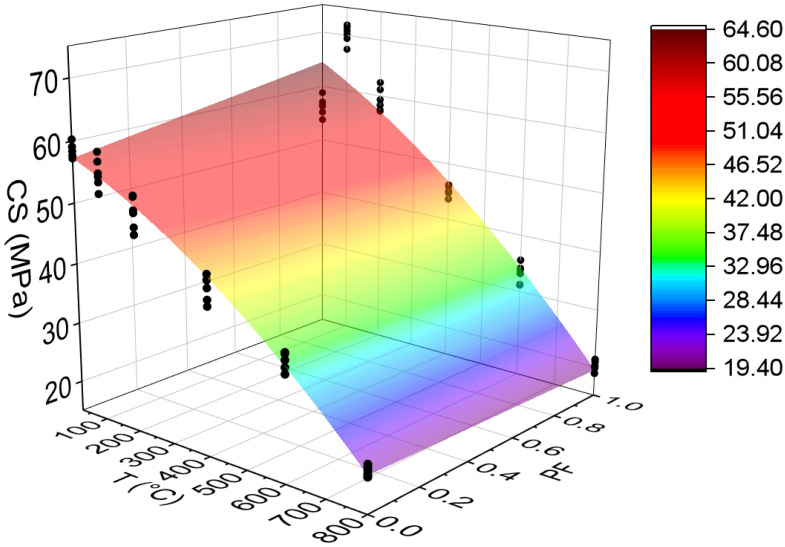
Quadratic multiple regression model fitting for compressive strength estimation.

**Figure 8 materials-15-04711-f008:**
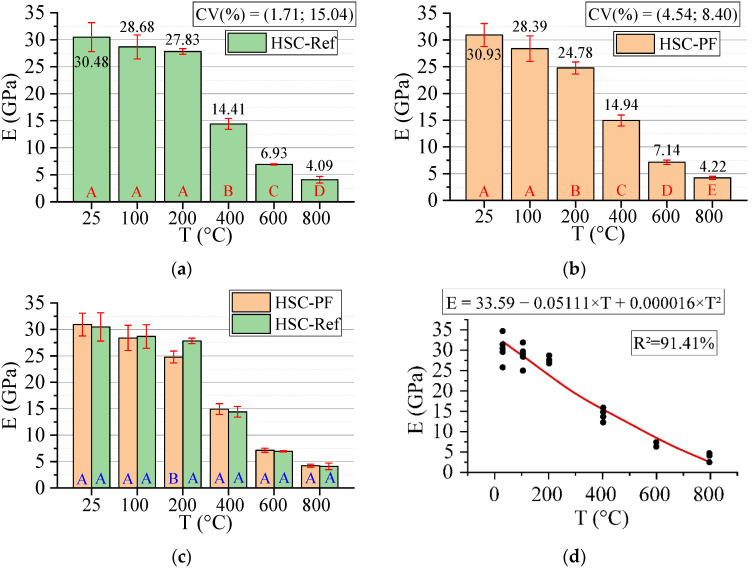
Elastic modulus in compression (E) results summary: (**a**) Tukey’s test for the samples without polypropylene fibers, (**b**) Tukey’s test for the samples with polypropylene fibers, (**c**) Tukey’s test for the samples without and with polypropylene fibers and (**d**) Quadratic regression model fit to the experimental data set considering only the effect of temperature as an independent factor. The uppercase letters A–E in figure mean equivalence between the sample results.

**Figure 9 materials-15-04711-f009:**
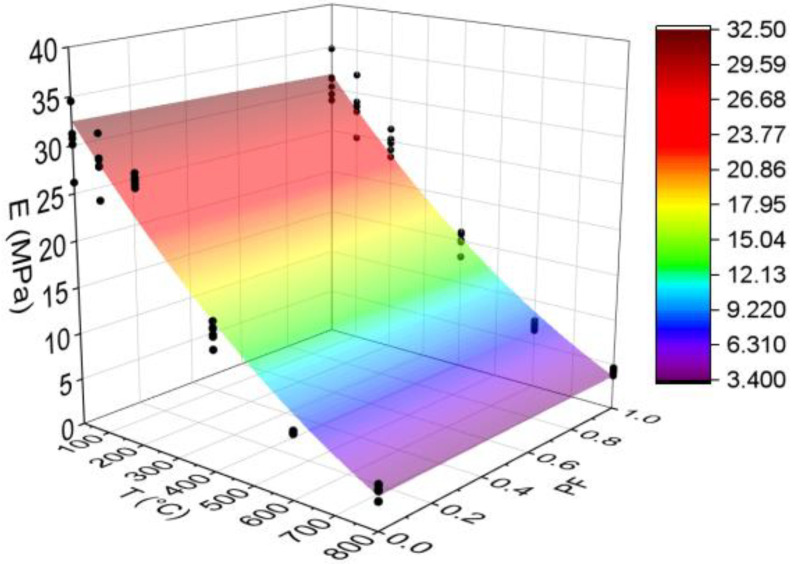
Quadratic multiple regression model fitting for elastic modulus estimation.

**Figure 10 materials-15-04711-f010:**
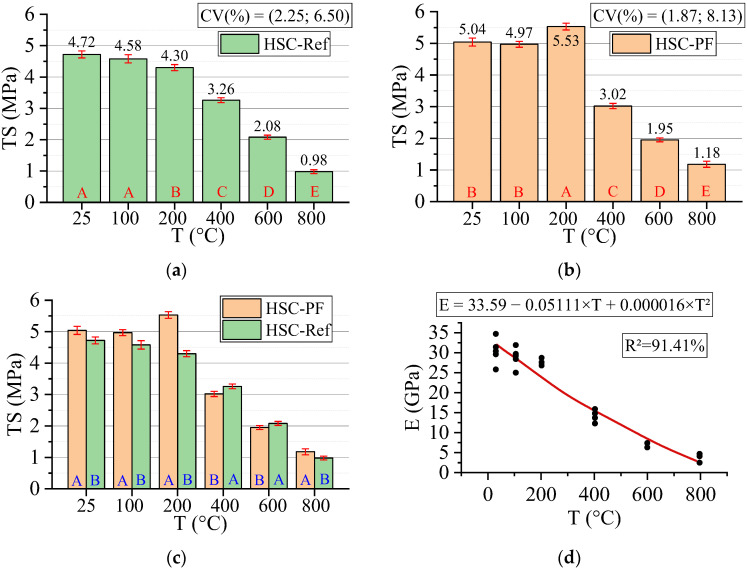
Summary of tensile strength (TS) results: (**a**) Tukey’s test for the samples without polypropylene fibers, (**b**) Tukey’s test for the samples with polypropylene fibers, (**c**) Tukey’s test for the samples without and with polypropylene fibers and (**d**) Quadratic regression model fit to the experimental data set considering only the effect of temperature as an independent factor. The uppercase letters A–E in figure mean equivalence between the sample results.

**Figure 11 materials-15-04711-f011:**
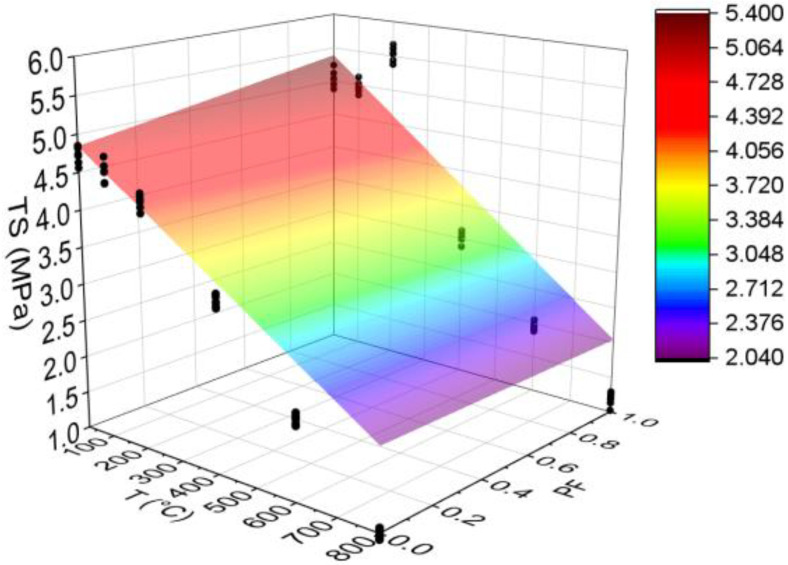
Quadratic multiple regression model fitting for tensile strength estimation.

**Figure 12 materials-15-04711-f012:**
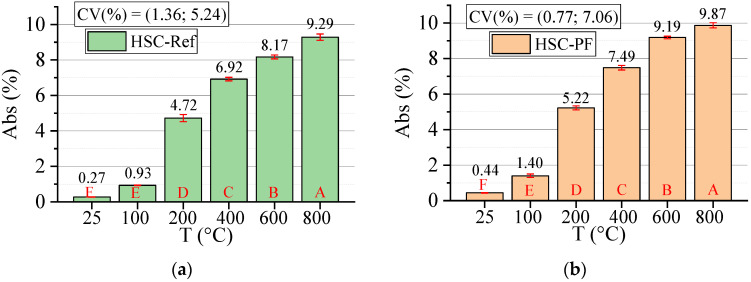
Summary of absorption (Abs) results: (**a**) Tukey’s test for the samples without polypropylene fibers, (**b**) Tukey’s test for the samples with polypropylene fibers, (**c**) Tukey’s test for the samples without and with polypropylene fibers and (**d**) Quadratic regression model fit to the experimental data set considering only the effect of temperature as an independent factor. The uppercase letters A–F in figure mean equivalence between the sample results.

**Figure 13 materials-15-04711-f013:**
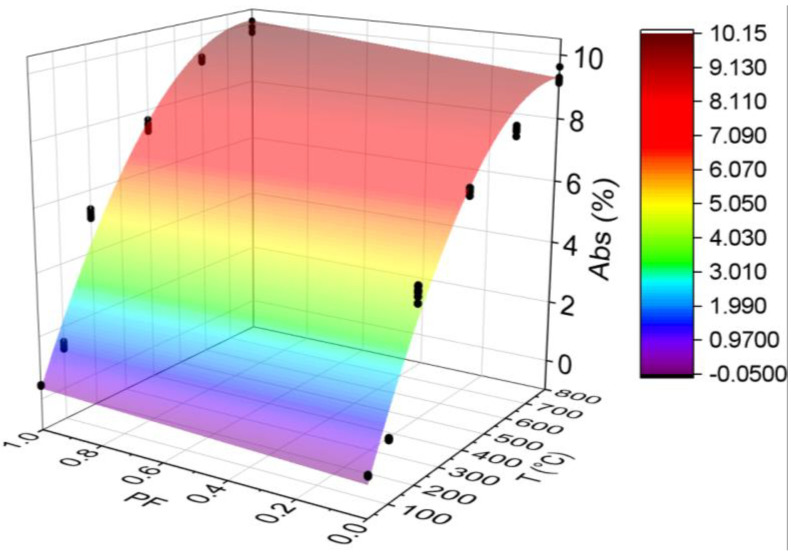
Quadratic multiple regression model fitting for absorption estimation.

**Figure 14 materials-15-04711-f014:**
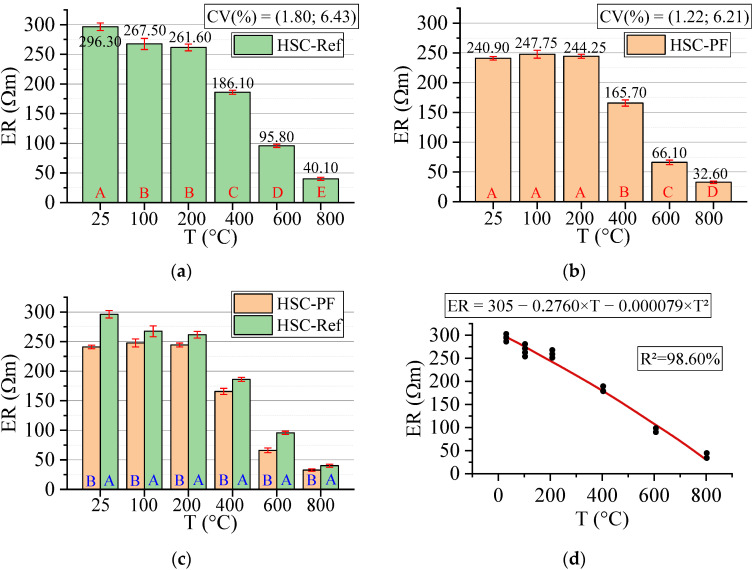
Summary of electrical resistivity (ER) results: (**a**) Tukey’s test for the samples without polypropylene fibers, (**b**) Tukey’s test for the samples with polypropylene fibers, (**c**) Tukey’s test for the samples without and with polypropylene fibers and (**d**) Quadratic regression model fit to the experimental data set considering only the effect of temperature as an independent factor. The uppercase letters A–E in figure mean equivalence between the sample results.

**Figure 15 materials-15-04711-f015:**
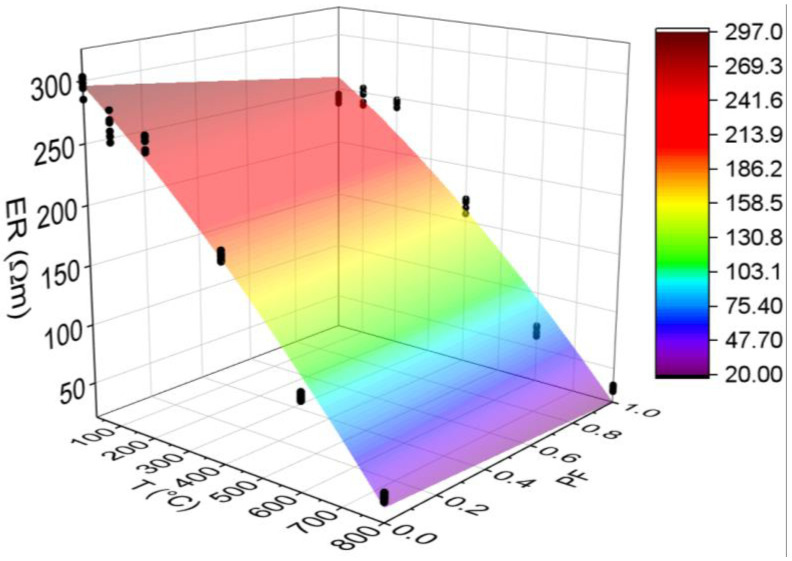
Quadratic multiple regression model fitting for electrical resistivity estimation.

**Figure 16 materials-15-04711-f016:**
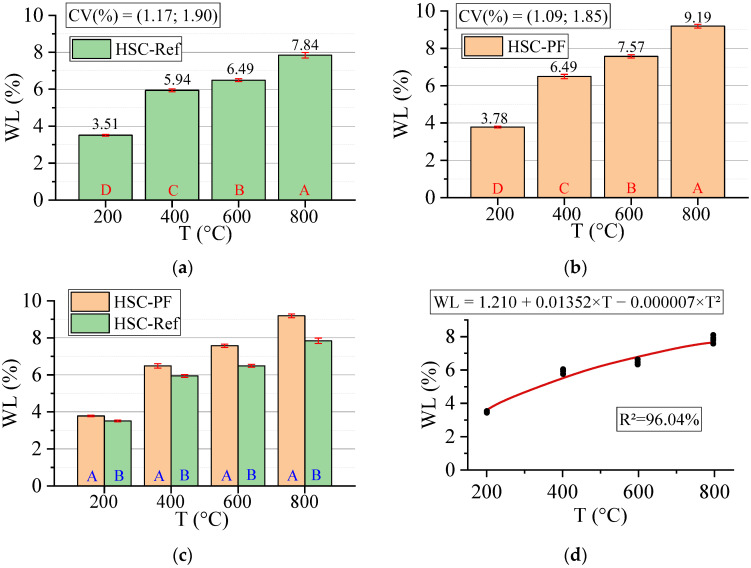
Summary of weight loss (WL) results: (**a**) Tukey’s test for the samples without polypropylene fibers, (**b**) Tukey’s test for the samples with polypropylene fibers, (**c**) Tukey’s test for the samples without and with polypropylene fibers and (**d**) Quadratic regression model fit to the experimental data set considering only the effect of temperature as an independent factor. The uppercase letters A–D in figure mean equivalence between the sample results.

**Figure 17 materials-15-04711-f017:**
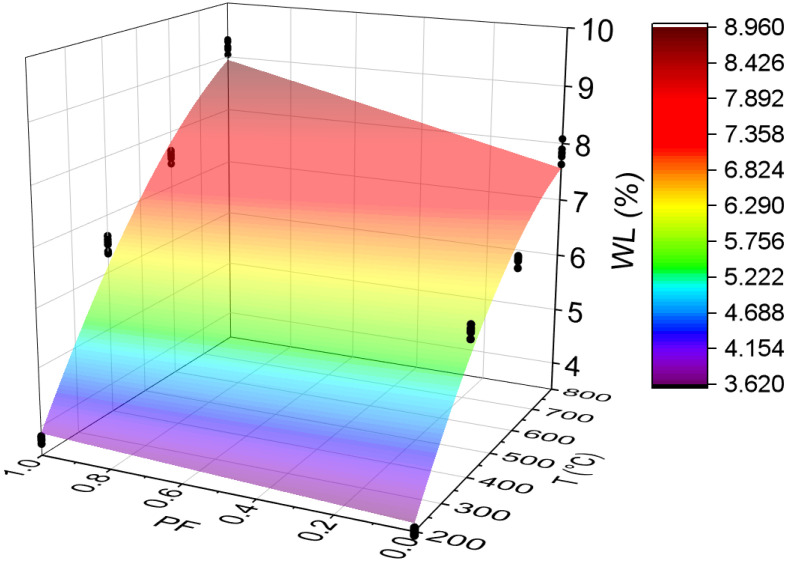
Quadratic multiple regression model fitting for weight loss estimation.

**Figure 18 materials-15-04711-f018:**
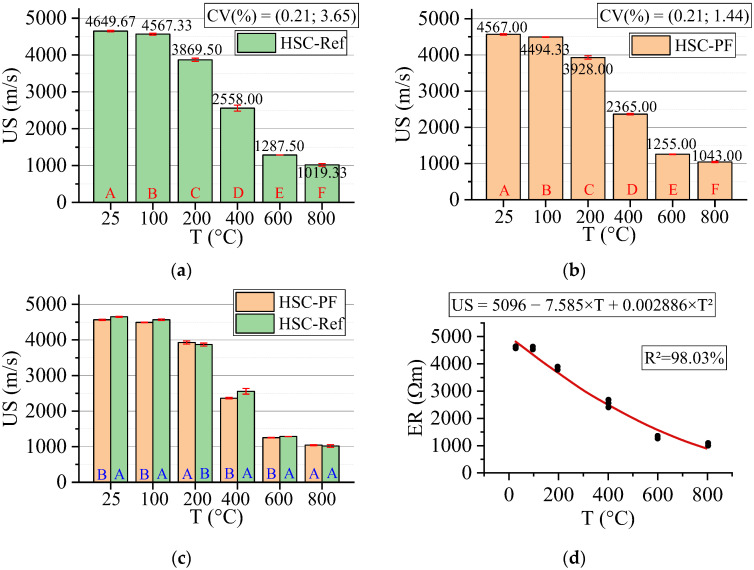
Summary of ultrasonic speed (US) results: (**a**) Tukey’s test for the samples without polypropylene fibers, (**b**) Tukey’s test for the samples with polypropylene fibers, (**c**) Tukey’s test for the samples without and with polypropylene fibers and (**d**) Quadratic regression model fit to the experimental data set considering only the effect of temperature as an independent factor. The uppercase letters A–F in figure mean equivalence between the sample results.

**Figure 19 materials-15-04711-f019:**
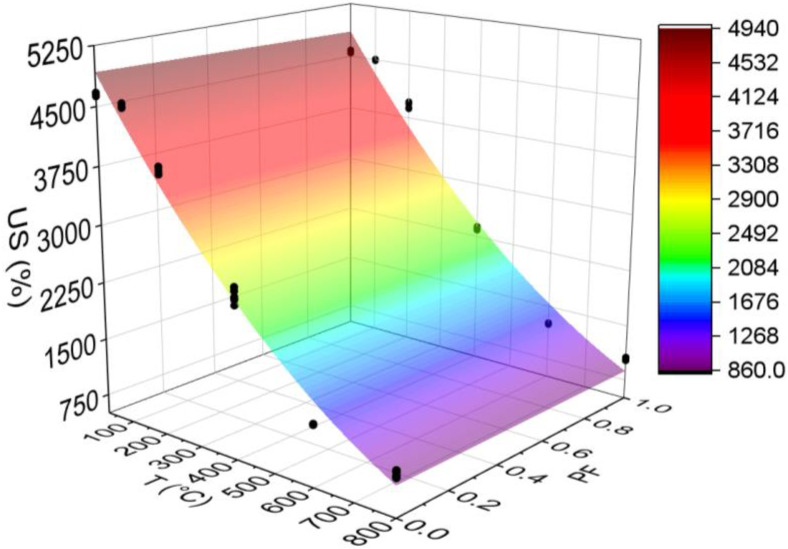
Quadratic multiple regression model fitting for ultrasonic velocity estimation.

**Figure 20 materials-15-04711-f020:**
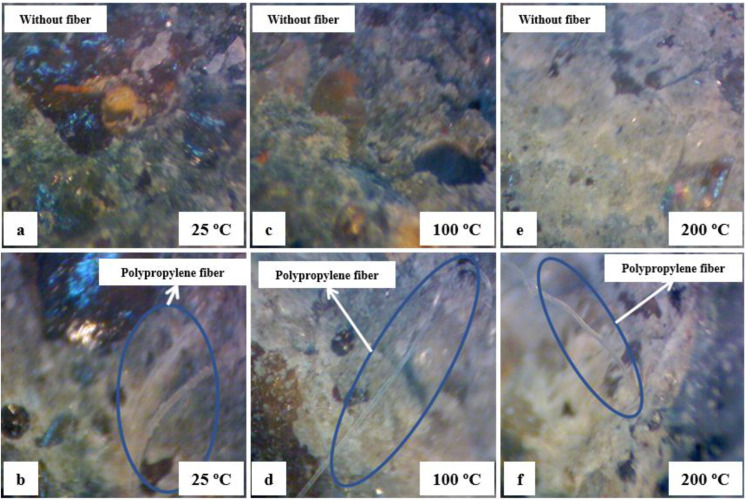
Microstructure of HSC: (**a**) 25 °C without fiber, (**b**) 25 °C with fiber, (**c**) 100 °C without fiber, (**d**) 100 °C with fiber, (**e**) 200 °C without fiber and (**f**) 200 °C with fiber.

**Figure 21 materials-15-04711-f021:**
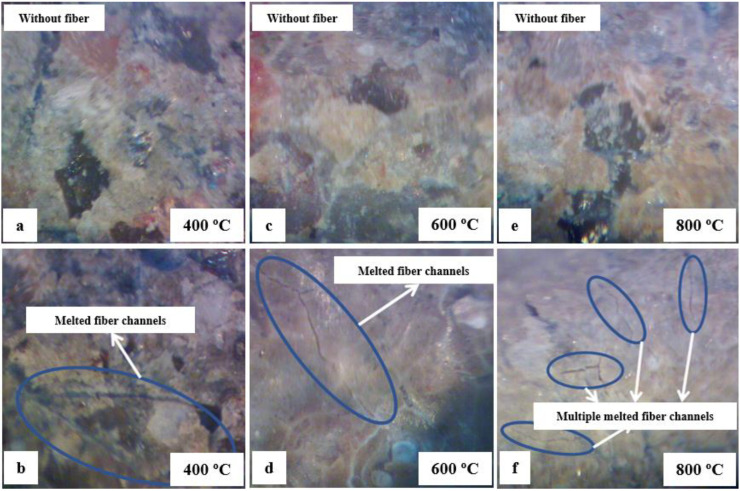
Microstructure of HSC: (**a**) 400 °C without fiber, (**b**) 400 °C with fiber, (**c**) 600 °C without fiber, (**d**) 600 °C with fiber, (**e**) 800 °C without fiber and (**f**) 800 °C with fiber.

**Table 1 materials-15-04711-t001:** Experimental treatments.

Tr	T (°C)	PF
1	25	Without
2	25	With
3	100	Without
4	100	With
5	200	Without
6	200	With
7	400	Without
8	400	With
9	600	Without
10	600	With
11	800	Without
12	800	With

**Table 2 materials-15-04711-t002:** Corrosion risk as a function of concrete surface electrical resistivity [62].

ER (ohm.m)	Corrosion Risk
ER > 200	Insignificant
100 < ER < 200	Low
50 < ER < 100	High
<50	Very high

**Table 3 materials-15-04711-t003:** Criteria for evaluating concrete based on ultrasonic speed at 28 days [65,66,67].

US (m/s)	Concrete Quality
US > 4.500	Excellent
3.500 < US < 4.500	Very good
3.000 < US < 3.500	Good
2.000 < US < 3.000	Regular
US < 2.000	Poor

## Data Availability

All the data in the tests of this study have been listed in the paper.

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
