# Peer review of "Residual Mechanical Properties and Durability of High-Strength Concrete with Polypropylene Fibers in High Temperatures"

_materials, 2022, doi:10.3390/ma15134711_

Round 1

Reviewer 1 Report

This study investigated the mechanical properties and durability of high-strength concrete containing polypropylene fibres exposed to high temperatures. This article needs some improvements to enhance its quality. The comments are listed below.

 1. Abstract: It would be better to mention the temperature in this form (25, 100, 200, 400, 600 and 800°C). Include the key finding of the results. The abstract should be revised based on these points.

2. Lines 47-58: Literatures highlighted only the study investigated and findings from the research are lacking. I suggest adding the key findings from each literature.

3. HSC specimens were exposed up to 800°C in this study, but the melting point of the polypropylene fibers is 170°C is mentioned in line 60. What are the benefits of polypropylene fibers since the melting point is approximately 170°C?.

4. Add a few literatures about the mechanical and durability properties of HSC.

5. At the end of the introduction, add a few lines about the novelty of the research.

6. Section 2.1 included the details of specific gravity, fineness, setting time of cement. Also, specific gravity, fineness modulus of fine aggregate. Add the details of gravel.

7. Section 2.1: mention the dosage of water reducer.

8. Section 2.1: mention the tensile strength of fibre used in this study.

9. Line 107: Check the unit “Two batches of cylindrical specimens measuring 10×20 cm2”

10. On what basis the dosage of polypropylene fibers (0.35 wt% of the cement mass) were selected?

11. Line 134: I suggest modifying this sentence “The specimens were subjected to tests inside the oven at” All specimens were exposed to 100, 200, 400, 600 and 800°C.

12. Mention the model parameters (Example β1, β2,……)

13. Line 206, 209 and 211 mention the figure number

14. Figures 5 (a) and (b), please use the dot to mention compressive strength (For example, 58.41 and not 58,41). Make similar corrections wherever applicable.

15. Equations 4-10: these equations are derived based on the experimental data. Are these equations applicable to global data? The validity of the equations is lacking?

16. How do you ensure the accuracy of the proposed equations?

Author Response

Response to Reviewer 1 Comments

Point 1: Abstract: It would be better to mention the temperature in this form (25, 100, 200, 400, 600 and 800°C). Include the key finding of the results. The abstract should be revised based on these points.

Response 1: the temperatures used in the experiment were added in the abstract, and one of the main results obtained was added. In order not to make the abstract too long, it was decided not to add any more results.

Point 2: Lines 47-58: Literatures highlighted only the study investigated and findings from the research are lacking. I suggest adding the key findings from each literature.

Response 2: one of the main findings of each research study has been added in the introduction. Other results have not been added to avoid making the introduction long and tiresome.

Point 3: HSC specimens were exposed up to 800°C in this study, but the melting point of the polypropylene fibers is 170°C is mentioned in line 60. What are the benefits of polypropylene fibers since the melting point is approximately 170°C?

Response 3: after this sentence, is explaining that the behavior of concrete with fibers after this temperature is not yet consolidated in the literature. As shown, two references (Kalifa et al. and Khoury) suggest two possible behaviors of concrete after this melting of fibers. Therefore, it is of utmost importance to analyze whether or not the fibers contribute at temperatures above 170°C.

Point 4: Add a few literatures about the mechanical and durability properties of HSC.

Response 4: as requested, in lines 47-58 a few properties obtained from the literature have been added. I believe we do not need to add this part anymore.

Point 5: At the end of the introduction, add a few lines about the novelty of the research.

Response 5: the main contribution of this research was added, which is to analyze the influence of polypropylene fibers on the HSC when requested at high temperatures, since it is not found research that has done this type of study. Furthermore, equations have been developed to predict the behavior of concrete as a function of temperature, obviating the need for experimental testing.

Point 6: Section 2.1 included the details of specific gravity, fineness, setting time of cement. Also, specific gravity, fineness modulus of fine aggregate. Add the details of gravel.

Response 6: additional information has been added.

Point 7: Section 2.1: mention the dosage of water reducer.

Response 7: additional information has been added.

Point 8: Section 2.1: mention the tensile strength of fibre used in this study.

Response 8: additional information has been added.

Point 9: Line 107: Check the unit “Two batches of cylindrical specimens measuring 10×20 cm2”.

Response 9: corrected.

Point 10: On what basis the dosage of polypropylene fibers (0.35 wt% of the cement mass) were selected?

Response 10: additional information has been added.

Point 11: Line 134: I suggest modifying this sentence “The specimens were subjected to tests inside the oven at” All specimens were exposed to 100, 200, 400, 600 and 800°C.

Response 11: done.

Point 12: Mention the model parameters (Example β1, β2,……)

Response 12: it is written in the text that βi are the model parameters.

Point 13: Line 206, 209 and 211 mention the figure number.

Response 13: in this case means in general: for all the next figures, what the letters of each one represents. This is the reason we didn't put the numbering of them. Therefore, it was put in another way to make it clearer.

Point 14: Figures 5 (a) and (b), please use the dot to mention compressive strength (For example, 58.41 and not 58,41). Make similar corrections wherever applicable.

Response 14: done for all figures, not only figure 5.

Point 15: Equations 4-10: these equations are derived based on the experimental data. Are these equations applicable to global data? The validity of the equations is lacking?

Response 15: for concrete that will be developed with the same fiber percentage as in this research or without fiber, all equations can be used to predict the properties of HSC with polypropylene fibers at high temperatures.

Point 16: How do you ensure the accuracy of the proposed equations?

Response 16: as stated, the accuracy of these equations is measured by the R². The closer to 1 (100%), the better the accuracy. To make it clearer, it has been added in the methodology.

Reviewer 2 Report

In the Reviewer opinion the research paper entitled “Residual mechanical properties and durability of high strength concrete with polypropylene fibers in high temperatures” is very good.

This work describes the influence of polypropylene fibers on the mechanical properties and durability of HSC at high temperatures. HSC specimens with 2 kg/m³ composed of polypropylene fibers are tested in a temperature range of 25 to 800ºC, followed by microstructural analysis. In addition, a statistical analysis is designed to identify the effect of factors, namely temperature and polypropylene fibers, and their interactions on mechanical properties and water absorption, electrical resistivity, mass loss and ultrasonic velocity.

Some comments which greatly enhance the understanding of the paper and its value are presented below. Specific issues that require further consideration are:

  1. The title of the manuscript is matched to its content.
  2. The Introduction generally covers the cases.
  3. The methodology was clearly presented.
  4. In the Reviewer’s opinion, the current state of knowledge relating to the manuscript topic has been presented.
  5. Experimental program and results looks interesting and was clearly presented.
  6. In the Reviewer’s opinion, the bibliography, comprising 61 references, is representative.
  7. An analysis of the manuscript content and the References shows that the manuscript under review constitutes a summary of the Author(s) achievements in the field.
  8. In the Reviewer’s opinion the manuscript is well written, and it should be published in the journal.

Author Response

Response to Reviewer 2 Comments

It did not request corrections.

Reviewer 3 Report

The submitted topic deals with the interesting and serious issue of fire safety of HSC. Basically, the paper has good potential to be published. However, there some issues that must be addressed and the manuscript modified and corrected.

i)                 Lack of novelty – is not clear what the paper gives new. There are several papers already published aimed at the similar research.

ii)                Spalling is caused by high temperature increase combined with water vapor release. In this paper, the heating rate was 1 °C/min which is slow. I don’t understand why standard ISO fire curve was not used in samples heating.

iii)               Description of conducted tests is completely missing. It must be completed.

iv)               At least density of prepared samples should be presented.

Some minor comments:

In section Conclusions, paragraph 2, use residual mechanical properties rather than residual strength mechanical properties.

Line 64 – vapor pressure release.

Author Response

Response to Reviewer 3 Comments

Point 1: Lack of novelty – is not clear what the paper gives new. There are several papers already published aimed at the similar research.

Response 1: the main contributions are: fiber behavior at high temperatures and the development of equations that predict the behavior of HSC at high temperatures. As requested by reviewer 1, the text was improved in this sense.

Point 2: Spalling is caused by high temperature increase combined with water vapor release. In this paper, the heating rate was 1°C/min which is slow. I don’t understand why standard ISO fire curve was not used in samples heating.

Response 2: the present research followed the premises of RILEM-TC-129 "Test Methods for Mechanical Properties of Concrete at High Temperatures". Therefore, this methodology is also valid and provides reliable results.

Point 3: Description of conducted tests is completely missing. It must be completed.

Response 3: the other reviewers said that the methodology is well presented and did not suggest any corrections. What information is missing? We are at your disposal to correct it and make your article even more relevant.

Point 4: At least density of prepared samples should be presented.

Response 4: additional information has been added.

Point 5: In section Conclusions, paragraph 2, use residual mechanical properties rather than residual strength mechanical properties.

Response 5: corrected.

Point 6: Line 64 – vapor pressure release.

Response 6: corrected.

Round 2

Reviewer 1 Report

All comments are addressed adequately.

Author Response

Response to Reviewer 1 Comments

It did not request corrections.

Reviewer 3 Report

Information on measurement techniques used for the assessment of compressive strength, fleural strength and elastic modulus must be provided, such as used equipement, loading force, experimental setup, etc. What standars were followed?

Author Response

Response to Reviewer 3 Comments

Point 1: Information on measurement techniques used for the assessment of compressive strength, fleural strength and elastic modulus must be provided, such as used equipement, loading force, experimental setup, etc. What standars were followed?

Response 1: The compressive strength test of the samples were carried out according to NBR 5739 [1] recommendations. EMIC hydraulic press PC 2000 CS model with 2000 kN capacity was used, at a loading rate of 0,45 ± 0,15 MPa/s. The equipment was calibrated according to NBR ISO 7500-1 [2]. To determine the elastic modulus in compression were carried out in accordance with the recommendations of NBR 8255-1 [3], using the hydraulic press previously mentioned, at a loading rate of 0,45 ± 0,15 MPa/s. The recommendations of NBR 7222 [4] were followed to determine the tensile strength, using the previously mentioned hydraulic press with a loading rate of 0,05 ± 0,01 MPa/s. All this data has been inserted in the manuscript in a new section (2.3) as well as the figures with the test schematics (Figure 4).

References

  1. ABNT NBR 5739 Concrete - Compression Test of Cylindrical Specimens. ABNT 2018, 9.
  2. ABNT NBR ISO 7500-1 Metallic Materials - Calibration and Verification of Static Uniaxial Testing Machines Part 1: Tension/Compression Testing Machines - Calibration and Verification of the Force-Measuring System. ABNT 2016, 19.
  3. ABNT NBR 8522-1 Hardened Concrete - Determination of Elasticity and Deformation Modulus Part 1: Static Modulus by Compression. ABNT 2021, 24.
  4. ABNT NBR 7222 Concrete and Mortar - Determination of the Tension Strength by Diametrical Compression of Cylindrical Test Specimens. ABNT 2011, 5.
